# Fault Tolerant Control of Quadrotor Based on Sensor Fault Diagnosis and Recovery Information

**Sunan Huang** *,†, **Fang Liao** † and **Rodney Swee Huat Teo**

Temasek Laboratories, National University of Singapore, T-Lab Building, 5A, Engineering Drive 1, Unit 09-02, Singapore 117411, Singapore

\* Correspondence: elehsn@gmail.com; Tel.: +65-98678943

† These authors contributed equally to this work.

**Abstract:** Drones have been developed for more than two decades. They have become central to the functions of various civil aviation and military services. Commercial usage of drones continues to grow steadily. As the drones have been used widely in different areas, this raises a safety concern, i.e., all the multi-rotors have an increased risk of motor or sensor faults. This paper considers a fault-tolerant control (FTC) problem against the inertial motion unit (IMU) sensor fault. First, a neural network estimator is built for the purpose of fault diagnosis. Second, a fault detection scheme is designed by comparing the IMU reading with the estimator, where it uses a logic rule to monitor the IMU state. Third, if the IMU sensor is in faulty state, the Euler angle estimator with neural network built is used to recover the IMU information which is fed into the controller designed. Finally, simulation studies are given to illustrate the effectiveness of the proposed FTC.

**Keywords:** fault diagnosis; neural networks; sensor fault; unmanned aerial vehicle



## 1. Introduction

Unmanned aerial vehicles (UAVs) have been used in many fields, such as photographing, monitoring traffic congestion, survellience, and multi-cameras coverage etc [1–3]. The missions for UAVs are becoming more and more challenging. An important problem in a UAV control system is to monitor the sensor or rotor changes when they are working. Much work has been done in this area. It has been shown that the use of an analytic model can allow early detection by measuring available variables. For example, in [4], the authors develop an incipient fault diagnosis method for a class of induction motors against stator/rotor winding faults; in [5], the authors present a incipient fault detection filter based on the generalised correntropy criterion; in [6], the authors propose a diagnosis method for broken rotor by using the analytic equations related to current signals; in [7], the authors present a fault detection method for a class of linear discrete time-varying systems. On the other hand, for UAV applications, it is important not only to detect sensor faults such as GPS, and inertial measurement units (IMU), but also to accommodate the faults (this is the so-called fault tolerant control). This topic has attracted increasing attention [8,9]. In this paper, IMU sensor fault is considered. The problem can be defined as a fault tolerant control that can accommodate IMU sensor faults, maintaining an acceptable performance. In [10], the authors presented a data-driven sensor fault diagnosis method. In [11], the authors also use data-driven method for detecting UAV faults. In [12], the authors propose a classifier for detecting UAV sensor fault. In [13], the authors use artificial intelligence (AI) method to learn the samples and estimate roll rate after the gyroscope has a fault. In [14], the authors develop an Euler angle estimator for improving the result of [13].

In this paper, we propose a fault-tolerant control to handle IMU sensor fault for quadrotor. The main idea is to build an angular rate estimator for the IMU sensor fault detection, which is also used for the IMU recovery. It should be noticed that the IMU fault affects not only angular rate but also the Euler angles. Our paper considers both

when designing the estimator. First, the observability analysis is used for selecting the variables for building the angular rate estimator from the sixteen states of the quadrotor studied. Second, the proposed estimator is composed of a long short-term memory (LSTM) neural network trained from the collected data. Third, an Euler angle estimator is designed for improving the angular rate estimation. Finally, a fault diagnosis and IMU recovery mechanism is proposed. The proposed method extends the result of [14] to more fields: (1) the variable selection of the proposed neural network observer is analyzed; (2) the convergence analysis of the proposed scheme is discussed; (3) the modified estimator of Euler angle is given. The contributions of the present paper include:

- Fault-tolerant control method for dealing with IMU sensor faults completely
- Neural network learning for estimating Euler angles and recovering IMU sensor information

## 2. Model of Quadrotor and IMU Sensor Fault

We use a quadrotor model as shown in [14] for our research and conducting simulation. The drone is assumed to be a rigid body, as shown in Figure 1.

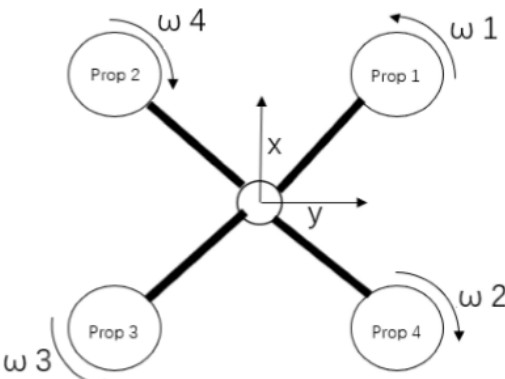

**Figure 1.** Quadrotor in cross (x) configuration.

For sensor faults, we consider roll or pitch faults caused by gyroscope sensor.

## 3. Framework of FTC for Sensor Detection and Recovery

In this section, we present a FTC framework against sensor faults. As shown in Figure 2, it is composed of a normal controller and a sensor processing unit. The former one works as a normal controller of quadrotor, and the latter one can monitor the system and recover the sensor information if the fault is diagnosed. The whole system is a fault-tolerant controller which can detect the IMU sensor faults and continue to maintain the drone to work safely by recovering the sensor by using the sensor estimator. It is observed that the two parts are separated and we can design each part independently.

We assume that the other sensors work well, for example compass, barometer and accelerometer.

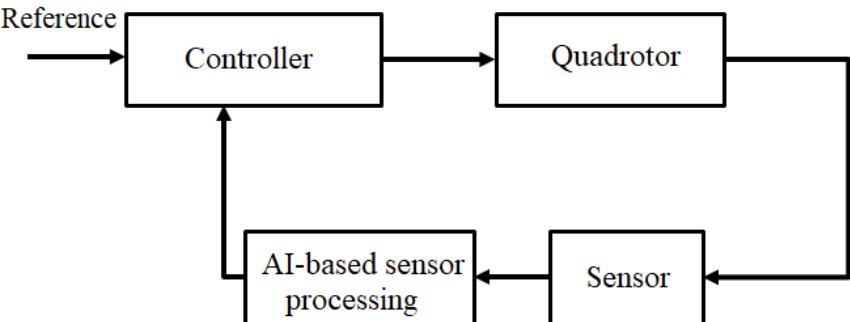

**Figure 2.** Architecture Overview of FTC.

## 4. AI-Based FTC against IMU Sensor Fault

As shown in the quadrotor model, the attitude state includes Euler angles and angular rates. These attitude states can be measured by IMU which contains accelerometer and gyroscope. Sometimes a gyroscope may have faults. This implies that the angular rates may not be available for the rate closed control loop. In this case, the drone may be out of control, causing the drone crash, even if the accelerometer is still available. Of course, in general, the compass is also available (this implies that $\psi$ is known). Thus, we intend to estimate the angular rates by observing other states such as the position (x, y, z), Euler angles as well as the control inputs. The designed estimator will be used for IMU fault detection as well as IMU recovery.

We will first discuss the analysis of the observability in order to design the estimator.

### 4.1. Selecting Variables

In a quadrotor model, we have the sixteen states including the positions, velocities, Euler angles, angular rates and four PWM signals. How many variables used for designing estimator angular rates is challenging. One way is to analyze nonlinear model. In [14], we have given the detailed model which is nonlinear. We can use nonlinear observability analysis to decide which variables will be used for the estimator design. The following theorem will be used in our analysis.

**Theorem 1.** *[15] Let*

$$\dot{x} = f(x, u) \tag{1}$$
$$y = h(x) \tag{2}$$

*where x is the manifold of dimension n. Let G be the set of all finite linear combinations formed with the Lie derivatives of $h_1, h_2, ...h_p$ with respect to $f$ and constant u. Let dG denote the set of the gradients of the elements of G. The system is weakly (locally) observable if the system satisfies the controllability rank condition at $x_0$, i.e., dG contains n linearly independent vectors.*

We focus on the attitude equations and have the following model.

$$\dot{\phi} = p + q\sin(\phi)\tan(\theta) + r\cos(\phi)\tan(\theta) \tag{3}$$

$$\dot{\theta} = q\cos(\phi) - r\sin(\phi) \tag{4}$$

$$\dot{\psi} = q\sin(\phi)\sec(\theta) + r\cos(\phi)\sec(\theta) \tag{5}$$

$$\dot{p} = \frac{M_p}{I_x} - qr\frac{(I_z - I_y)}{I_x} \tag{6}$$

$$\dot{q} = \frac{M_q}{I_y} - pr\frac{(I_x - I_z)}{I_y} \tag{7}$$

$$\dot{r} = \frac{M_r}{I_z} - pq\frac{(I_y - I_x)}{I_z}. \tag{8}$$

The angular rate controls $M_p$, $M_q$, and $M_r$ can be further allocated by the following equations

$$Mp = k_T l_y(-\omega_1^2 - \omega_2^2 + \omega_3^2 + \omega_4^2) \tag{9}$$

$$Mq = k_T l_x(\omega_1^2 - \omega_2^2 - \omega_3^2 + \omega_4^2) \tag{10}$$

$$Mr = k_D(\omega_1^2 - \omega_2^2 + \omega_3^2 - \omega_4^2) \tag{11}$$

$$Tz = k_T(\omega_1^2 + \omega_2^2 + \omega_3^2 + \omega_4^2) \tag{12}$$

where $\omega_i$ represents the spinning speed of the $i$th motor, $l_x$ and $l_y$ are the arm length along x and y axes, respectively, and $k_T$ and $k_D$ are the thrust and torque constants, respectively.

Furthermore, we assume that the Euler angles have small changes. This implies that $p \approx \dot{\phi}, q \approx \dot{\theta}, r \approx \dot{\psi}$. Rearranging the attitude equations, we have

$$
\begin{aligned}
\dot{x} &= f(x) + g(x, u) \\
&= \begin{bmatrix} x_2 + x_4\sin(x_1)\tan(x_3) + x_6\cos(x_1)\tan(x_3) \\ -x_4 x_6 \frac{(I_z - I_y)}{I_x} \\ x_4\cos(x_1) - x_6\sin(x_1); \\ -x_2 x_6 \frac{(I_x - I_z)}{I_y} \\ x_4\sin(x_1)\sec(x_3) + x_6\cos(x_1)\sec(x_3); \\ -x_2 x_4 \frac{(I_y - I_x)}{I_z} \end{bmatrix} + \begin{bmatrix} 0 \\ \frac{M_p}{I_x} \\ 0 \\ \frac{M_q}{I_y} \\ 0 \\ \frac{M_r}{I_z} \end{bmatrix}
\end{aligned} \tag{13}
$$

$$y = \begin{bmatrix} x_1 \\ x_3 \\ x_5 \end{bmatrix} = h(x) \tag{14}$$

where

$$x = [\phi, \dot{\phi}, \theta, \dot{\theta}, \psi, \dot{\psi}]^T. \tag{15}$$

The linear combinations with the Lie derivatives are given by

$$G = \begin{bmatrix} x_1 \\ x_3 \\ x_5 \\ x_2 \\ x_4 \\ x_6 \\ -x_4 x_6 \frac{(I_z - I_y)}{I_x} + \frac{M_{p0}}{I_x} \\ -x_2 x_6 \frac{(I_x - I_z)}{I_y} + \frac{M_{q0}}{I_y} \\ -x_2 x_4 \frac{(I_y - I_x)}{I_z} + \frac{M_{r0}}{I_z} \end{bmatrix} \tag{16}$$

where $\Omega_{r0}$ is a constant, and $M_{p0}, M_{q0}, M_{r0}$ are the control input constants.

The set of the gradients of G is given by

$$dG = \begin{bmatrix} 1 & 0 & 0 & 0 & 0 & 0 \\ 0 & 0 & 1 & 0 & 0 & 0 \\ 0 & 0 & 0 & 0 & 1 & 0 \\ 0 & 1 & 0 & 0 & 0 & 0 \\ 0 & 0 & 0 & 1 & 0 & 0 \\ 0 & 0 & 0 & 0 & 0 & 1 \\ 0 & 0 & 0 & -x_6\frac{(I_z - I_y)}{I_x} & 0 & -x_4\frac{(I_z - I_y)}{I_x} \\ 0 & -x_6\frac{(I_x - I_z)}{I_y} & 0 & 0 & 0 & -x_2\frac{(I_x - I_z)}{I_y} \\ 0 & -x_4\frac{(I_y - I_x)}{I_z} & 0 & x_2\frac{(I_y - I_x)}{I_z} & 0 & 0 \end{bmatrix} \tag{17}$$

Check the rank of $d$G. It is observed that rank($d$G)=6. This means that the system is locally observable at the hover state and is verified near hovering state. This property will be lost away from that point.

Furthermore, we consider a full description of quadrotor as

$$\left. \begin{array}{rcl} \ddot{\mathbf{x}} & = & \frac{1}{m}(R(\phi, \theta, \psi)\bar{T} - m\bar{g}) \\ \dot{\phi} & = & p + q\sin(\phi)\tan(\theta) + r\cos(\phi)\tan(\theta) \\ \dot{\theta} & = & q\cos(\phi) - r\sin(\phi) \\ \dot{\psi} & = & q\sin(\phi)\sec(\theta) + r\cos(\phi)\sec(\theta) \\ \dot{p} & = & \frac{M_p}{I_x} - qr\frac{(I_z - I_y)}{I_x} \\ \dot{q} & = & \frac{M_q}{I_y} - pr\frac{(I_x - I_z)}{I_y} \\ \dot{r} & = & \frac{M_r}{I_z} - pq\frac{(I_y - I_x)}{I_z} \end{array} \right\} \tag{18}$$

where **x** represents the position $[x, y, z]$, and $\bar{T} = [0; 0; T]^T$, $\bar{g} = [0; 0; g]^T$.

If considering the measured variables (velocities) to be $[\dot{x}, \dot{y}, \dot{z}]$, we check if the system is observable. Still, we check the rank of dG and it is not 9, but 7. Thus, it is concluded that in this situation, the system is not observable.

Based on the analysis above, we select ten variables related to the estimator design as shown in Table 1.

**Table 1.** Variables used for estimator design.

| Symbol | Parameter | Unit |
|--------|-----------|------|
| $\phi$ | Roll | rad |
| $\theta$ | Pitch | rad |
| $\psi$ | Yaw | rad |
| $p$ | Roll rate | rad/s |
| $q$ | Pitch rate | rad/s |
| $r$ | Yaw rate | rad/s |
| $\omega_1$ | Motor 1's speed | rad/s |
| $\omega_2$ | Motor 2's speed | rad/s |
| $\omega_3$ | Motor 3's speed | rad/s |
| $\omega_4$ | Motor 4's speed | rad/s |

### 4.2. Neural Network Angular Rate Estimator

For handling gyroscope fault, we will consider to use AI techniques to build a nonlinear mapping function between input and output by learning sample data sets. One of existing

AI methods is the LSTM neural network which is powerful in dealing with time series problems. The advantage of LSTM is that the current unit can get the information of all the units before this unit, but the disadvantage is that the information of the units after this unit cannot be obtained. For our application, it is to model sequential data and the LSTM is suitable for our modelling.

It should be noticed that the flight is a time dynamical behavior. Thus, the built LSTM model can be used for estimating or predicting the next state. The variables used in the network are attitude information–roll, pitch and yaw and their corresponding three angular rates. To consider the control action, each PWM signal (it is represented by propeller speed) is also considered. As shown by the observability computation, these ten variables will be used for the LSTM modelling.

It denotes

$$\begin{aligned} x &= [x_1, x_2, ...x_{10}]^T \\ &= [\phi, \theta, \psi, p, q, r, \omega_1^2, \omega_2^2, \omega_3^2, \omega_4^2]^T. \end{aligned} \tag{19}$$

For building the LSTM network, the following input and output vectors are used

$$X_t = \begin{bmatrix} x_{1,t-D+1}, & x_{1,t-D+2} & \cdots & x_{1,t} \\ x_{2,t-D+1}, & x_{2,t-D+2} & \cdots & x_{2,t} \\ \cdots & \cdots & \cdots & \cdots \\ x_{10,t-D+1}, & x_{10,t-D+2}, & \cdots & x_{10,t} \end{bmatrix}, \tag{20}$$

$$Y_t = \begin{bmatrix} x_{1,t+1} \\ x_{2,t+1} \\ \vdots \\ x_{6,t+1} \end{bmatrix} \tag{21}$$

where $X_t$ represents the input of the LSTM model at time $t$, $Y_t$ including Euler angles and angular rates, represents the LSTM output which is used for observing roll,pitch and yaw rates, and $D$ represents the data size. In practices, $D$ will be determined by observing the training performance.

As shown in Figure 3, in our application, the LSTM network consists of three parts: input,LSTM and output layers.

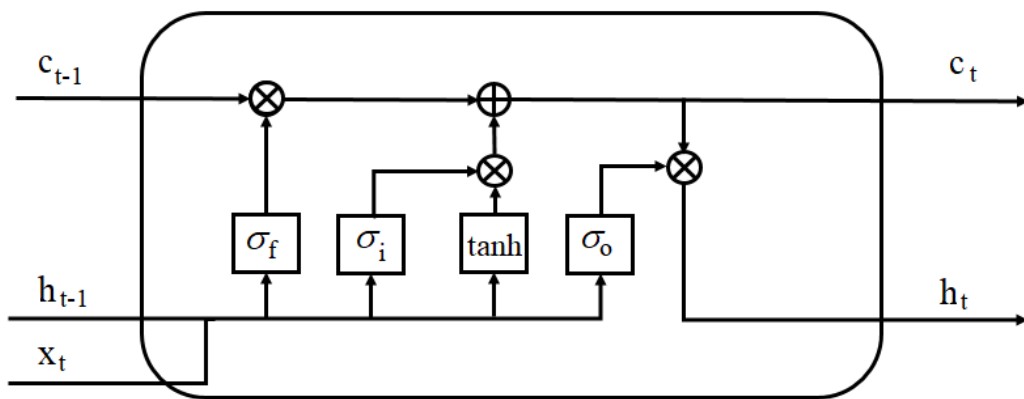

**Figure 3.** LSTM network used.

By the training, the angular rates can be estimated by using the LSTM network as show below

$$X_{k+1} \quad = \quad F_{NN}(X_k, X_{k-1}, ..., X_{k-d1}, u_k, ..., u_{k-d1})$$

where $X_k = [\phi(k), \theta(k), \psi(k), p(k), q(k), r(k)]^T$, $u_k = [\omega_1^2(k), \omega_2^2(k), \omega_3^2(k), \omega_4^2(k)]^T$, and $F_{NN}$ represents the LSTM network. The input and output of the model are shown in Figure 4.

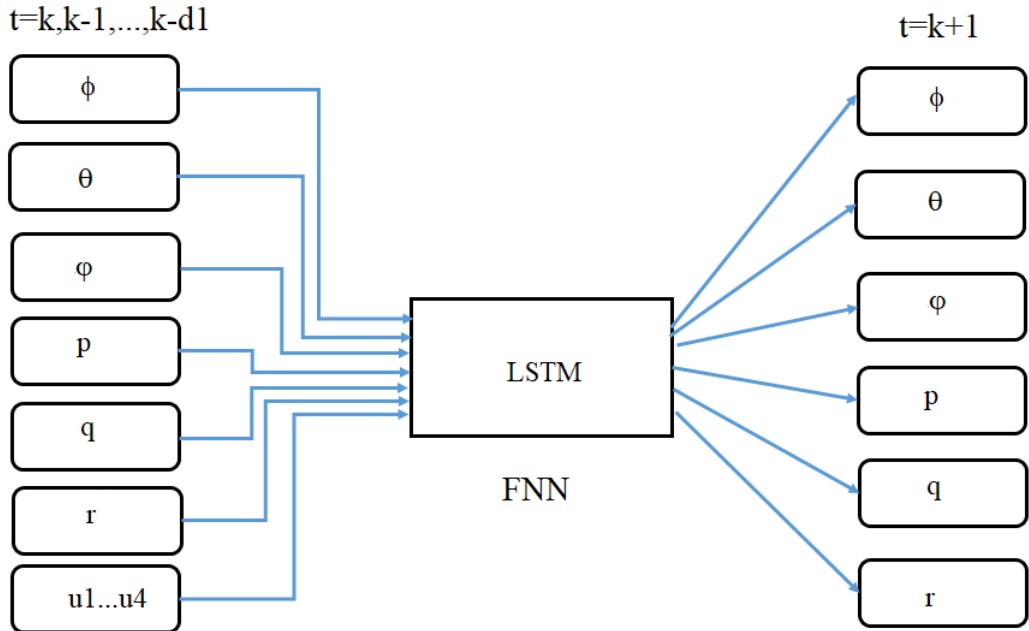

**Figure 4.** Modelling diagram of LSTM network.

Before the LSTM training, the data should be normalized for the purpose of the calculation stability. After that, the trained prediction model is used to denormalize the prediction result, so as to compare and analyze the error with the actual value. After the prediction sample data are constructed, the LSTM model can be established for training and prediction according to the following steps.

Step 1. Select a flight or simulation data, and construct training samples and prediction samples.

Step 2. Data preprocessing, normalize training samples and prediction samples.

Step 3. Input the training samples into the LSTM, change the number of hidden layer nodes of the LSTM, and determine the network structure by minimizing the error between the output result and the real sample.

Step 4. Input the prediction sample into the trained network, get the trajectory prediction result, and perform denormalization.

Step 5. compare with other algorithms and analyze the prediction effect. Therefore, the flow chart of the LSTM-based trajectory prediction model algorithm is shown in Figure 5.

The loss function uses the root mean square error (RMSE) function which is given by

$$E_{RMSE} = \sqrt{\frac{1}{N} \sum_{i=1}^{N} (y_{LSTM\ prediction} - y_i)^2} \tag{22}$$

where $y_{LSTM\ prediction}$ is the LSTM output, and $y_i$ is the output of the sample.

It is observed that the angular rate information can be obtained by

$$
\begin{aligned}
\hat{X}_{k+1} &= F_{NN}(\hat{X}_k, \hat{X}_{k-1}, ..., \hat{X}_{k-d1}, u_k, ..., u_{k-d1}) + L(Z_k - \hat{Z}_k) & (23)\\
\hat{Z}_k &= C\hat{X}_k & (24)
\end{aligned}
$$

where

$$
\begin{aligned}
\hat{X}_k &= [\hat{\phi}(k), \hat{\theta}(k), \hat{\psi}(k), \hat{p}(k), \hat{q}(k), \hat{r}(k)]^T, \\
u_k &= [\omega_1^2(k), \omega_2^2(k), \omega_3^2(k), \omega_4^2(k)]^T, \\
C &= \begin{bmatrix} 1 & 0 & 0 & 0 & 0 & 0 \\ 0 & 1 & 0 & 0 & 0 & 0 \\ 0 & 0 & 1 & 0 & 0 & 0 \end{bmatrix} \\
Z_k &= [\phi(k), \theta(k), \psi(k)]^T, \\
\hat{Z}_k &= [\hat{\phi}(k), \hat{\theta}(k), \hat{\psi}(k)]^T.
\end{aligned}
$$

For analysis purpose, we re-arrange the equation as

$$
\begin{aligned}
X_{k+1} &= X_k + F_{NN}(X_{k-1}, X_{k-2}, ..., X_{k-d1}, u_{k-1}, ..., u_{k-d1}) - X_k \\
&= X_k + \bar{F}_{NN}(X_{k-1}, X_{k-2}, ..., X_{k-d1}, u_{k-1}, ..., u_{k-d1})
\end{aligned}
$$

where $\bar{F}_{NN} = F_{NN} - X_k$. It should be noticed that this can also be written as

$$
\begin{aligned}
\hat{X}_{k+1} &= \hat{X}_k + \bar{F}_{NN}(\hat{X}_{k-1}, \hat{X}_{k-2}, ..., \hat{X}_{k-d1}, u_{k-1}, ..., u_{k-d1}) \\
&\quad + L(Z_k - \hat{Z}_k) \\
Z_k &= CX_k \\
\hat{Z}_k &= C\hat{X}_k
\end{aligned}
$$

Thus, we have the error equation

$$
\begin{aligned}
\tilde{X}_{k+1} &= \tilde{X}_k + \varepsilon(\cdot) - LC\tilde{X}_k \\
&= (I - LC)\tilde{X}_k + \varepsilon(\cdot) \\
&= \bar{A}\tilde{X}_k + \varepsilon(\cdot) & (25)\\
\tilde{Z}_k &= C\tilde{X}_k & (26)
\end{aligned}
$$

where $\tilde{X}_k = X_k - \hat{X}_k$, $\bar{A} = I - LC$,

$$
\begin{aligned}
\varepsilon(\cdot) &= \bar{F}_{NN}(X_{k-1}, ..., X_{k-d1}, ..., u_{k-d1}) \\
&\quad - \bar{F}_{NN}(\hat{X}_{k-1}, ..., \hat{X}_{k-d1}, u_{k-1}, ..., u_{k-d1}), & (27)
\end{aligned}
$$

and $\tilde{Z}_k = Z_k - \hat{Z}_k$. A reasonable assumption is that the output of the actual system is bounded. It should be noticed that the neural network output $\bar{F}$ is bounded. Thus, $\varepsilon$ is also bounded.

It can be seen that the proposed LSTM network is a nonlinear observer-like estimator by using the observed vector $Z_k$. Such a design can enhance the performance of the estimator. The convergence analysis of the proposed estimator can be given below.

Define a Lyapunov function

$$
V = \tilde{X}_k^T P \tilde{X}_k \tag{28}
$$

Thus, we have

$$
\begin{aligned}
\Delta V &= \tilde{X}_{k+1}^T P \tilde{X}_{k+1} - \tilde{X}_k^T P \tilde{X}_k \\
&= \tilde{X}_k^T (\bar{A}^T P \bar{A} - P) \tilde{X}_{k+1} + 2\tilde{X}_k^T \bar{A}^T P \varepsilon + \varepsilon^T P \varepsilon \\
&\leq -\lambda_{min}(Q)||\tilde{X}_k||^2 + \frac{\lambda_{min}}{2}(Q)||\tilde{X}_k||^2 \\
&\quad + \frac{2}{\lambda_{min}(Q)}||\bar{A}^T P||^2 \parallel \varepsilon \parallel^2 + \parallel P \parallel \parallel \varepsilon \parallel^2 \\
&= -\frac{\lambda_{min}}{2}(Q)||\tilde{X}_k||^2 + \bar{\varepsilon}
\end{aligned}
\tag{29}
$$

where we have used the inequality $2ab \leq \sigma a^2 + \frac{1}{\sigma}b^2$, $\bar{\varepsilon} = max\{\frac{2}{\lambda_{min}(Q)}||\bar{A}^T P||^2 \parallel \varepsilon \parallel^2 + \parallel P \parallel \parallel \varepsilon \parallel^2\}$, $Q$ can be obtained from the following equation

$$
\bar{A}^T P \bar{A} - P + Q = 0.
\tag{30}
$$

with $\lambda_{min}(Q)$ denoted as the minimum eigenvalue of $Q$. It should be noticed that $\Delta V < 0$ if $||\bar{X}_k|| > \sqrt{\frac{\bar{\varepsilon}}{\frac{\lambda_{min}}{2}(Q)}}$. This implies that it is uniformly ultimate bounded according to [16].

It can be observed from the above that the LSTM neural network is a key part in the proposed estimator. It can be trained by learning the samples collected from the drone.

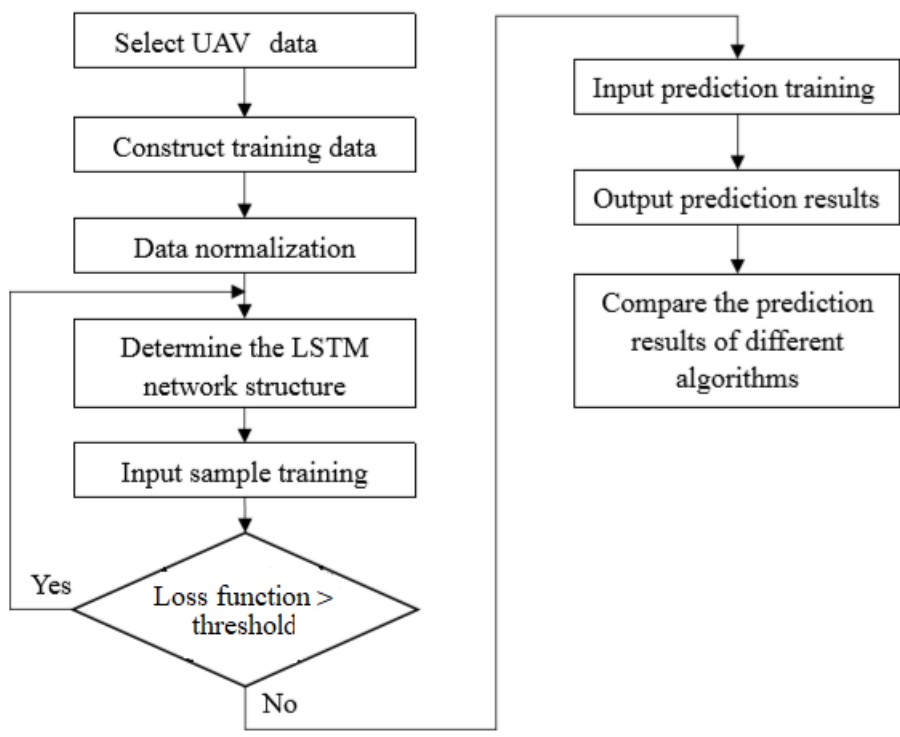

**Figure 5.** Algorithm flow chart.

### 4.3. Euler Angle Estimator

In a quadrotor, the gyroscope sensor not only affects the angular rate measurement, but also Euler angle reading. This implies that when the gyroscope is in faulty state, it also affects the Euler angle reading. Since we assume that the compass works well, the yaw angle is still fine. We will estimate the Euler angles $(\phi, \theta)$ when the gyroscope fault occurs.

For Euler angle estimation, we use the following translational motion equations.

$$\frac{m}{Tz}E_m = \frac{m}{Tz}\left[\begin{pmatrix} \ddot{x} \\ \ddot{y} \\ \ddot{z} \end{pmatrix} - \begin{pmatrix} 0 \\ 0 \\ g \end{pmatrix}\right]$$

$$= \begin{bmatrix} cos(\phi)sin(\theta)cos(\psi) + sin(\phi)sin(\psi) \\ cos(\phi)sin(\theta)sin(\psi) - sin(\phi)cos(\psi) \\ cos(\phi)cos(\theta) \end{bmatrix}$$

where $E_m = \mathbf{a}_I - \mathbf{g}$, $\mathbf{a}_I = [\ddot{x}, \ddot{y}, \ddot{z}]^T$ and $\mathbf{g} = [0, 0, g]^T$.

Let

$$b = \begin{bmatrix} cos(\phi)sin(\theta)cos(\psi) + sin(\phi)sin(\psi) \\ cos(\phi)sin(\theta)sin(\psi) - sin(\phi)cos(\psi) \\ cos(\phi)cos(\theta) \end{bmatrix}. \tag{31}$$

Taking normalization of two sides, we have

$$\frac{\frac{m}{Tz}E_m}{||\frac{m}{Tz}E_m||} = \frac{b}{||b||}. \tag{32}$$

It should be noticed that

$$||b|| = \sqrt{c^2\phi s^2\theta c^2\psi + s^2\phi s^2\psi + c^2\phi s^2\theta s^2\psi + s^2\phi c^2\psi + c^2\phi c^2\theta}$$

$$= \sqrt{c^2 s^2\theta + s^2 + c^2\phi c^2\theta}$$

$$= \sqrt{c^2 s + s^2} = 1$$

where $c^2\phi = cos^2(\phi), s^2\theta = sin^2(\theta), c^2\psi = cos^2(\psi), s^2\phi = sin^2(\phi), s^2\psi = sin^2(\psi), c^2\theta = cos^2(\theta)$. Thus, we hve

$$\frac{\frac{m}{Tz}E_m}{||\frac{m}{Tz}E_m||} \rightarrow E_n = \frac{E_m}{||E_m||} \tag{33}$$

$$E_n = \begin{bmatrix} cos(\phi)sin(\theta)cos(\psi) + sin(\phi)sin(\psi) \\ cos(\phi)sin(\theta)sin(\psi) - sin(\phi)cos(\psi) \\ cos(\phi)cos(\theta) \end{bmatrix}. \tag{34}$$

From our calculation, it can be seen that the vector $E_n = [e_{n1}, e_{n2}, e_{n3}]^T$ does not have the mass $m$ and thrust $Tz$. This simplifies our computation. Thus, we have the following equations

$$\hat{\phi} = sin^{-1}(-e_{n1}sin(\psi) + e_{n2}cos(\psi)) \tag{35}$$

$$\hat{\theta} = sin^{-1}\left(\frac{e_{n1}cos(\psi) + e_{n2}sin(\psi)}{\sqrt{e_{n3}^2 + (e_{n1}cos(\psi) + e_{n2}sin(\psi))^2}}\right). \tag{36}$$

The estimated Euler angle should be calibrated by using the measured acceleration readings $\phi_{accel}, \theta_{accel}$. The working principle of using acceleration to estimate Euler angle is as follows.

We can obtain the acceleration information $\mathbf{a}_b = [a_{bx}, a_{by}, a_{bz}]^T$ and the gravity vector $\mathbf{g}$. During the hovering state, by ignoring the effect of yaw, we have

$$\mathbf{a}_b = \begin{bmatrix} cos\phi & 0 & -sin\theta \\ sin\theta sin\phi & cos\phi & cos\theta sin\phi \\ sin\theta cos\phi & -sin\phi & cos\theta cos\phi \end{bmatrix} \mathbf{g} \tag{37}$$

Thus, we have

$$\phi_{accel} = arctan(\frac{a_{by}}{a_{bz}}) \tag{38}$$

$$\theta_{accel} = arcsin(\frac{a_{bx}}{g}) \tag{39}$$

The following calibration is used for estimating angles

$$\hat{\phi}_{est} = \alpha\hat{\phi} + (1 - \alpha)\phi_{accel} \tag{40}$$

$$\hat{\theta}_{est} = \alpha\hat{\theta} + (1 - \alpha)\theta_{accel} \tag{41}$$

where $\alpha$ is a factor which is a filter-like one. This factor is determined by training. The estimation of Euler angle is obtained.

### 4.4. Imu Fault Detection and Recovery Decision Mechanism

From the obtained LSTM neural network, we can obtain the estimated LSTM network error, that is $\tilde{y} = y - \hat{y}$, where $y$ is the sample data, and $\hat{y}$ is the network output. Based on the statistic information, we can obtain a threshold given by

$$Threshold = mean\ value + const * deviation \tag{42}$$

where *mean value* represents the mean of the statistic information, *deviation* represents the standard deviation of the statistic information, and *const* is determined by users. The fault diagnosis mechanism regarding IMU sensor is given by

- If the LSTM neural network estimated $||\tilde{y}|| > Threshold$, IMU has faults in angular rate;
- otherwise, IMU has no fault occurrence. In this situation, the actual angular rate measurement is used.

Once the angular rate fault in IMU sensor is detected, the recovery should be made by using the proposed estimator. Figure 6 shows the angular rate estimator. The entire fault-tolerant control against sensor fault with diagnosis and recovery scheme is shown in Figure 7, where the fault diagnosis determines the switching between the sensor reading and recovery.

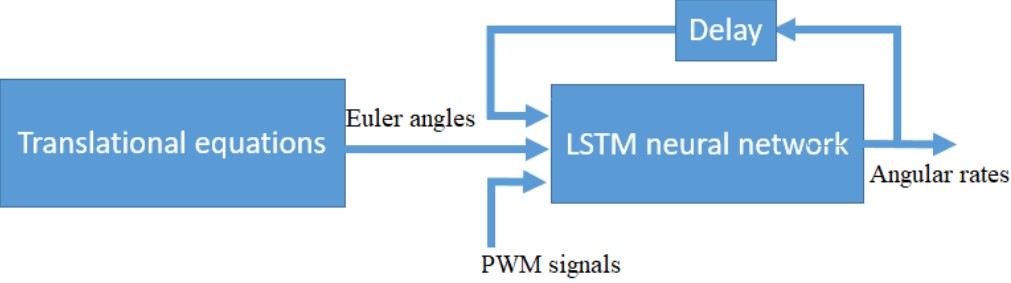

**Figure 6.** Sensor estimator.

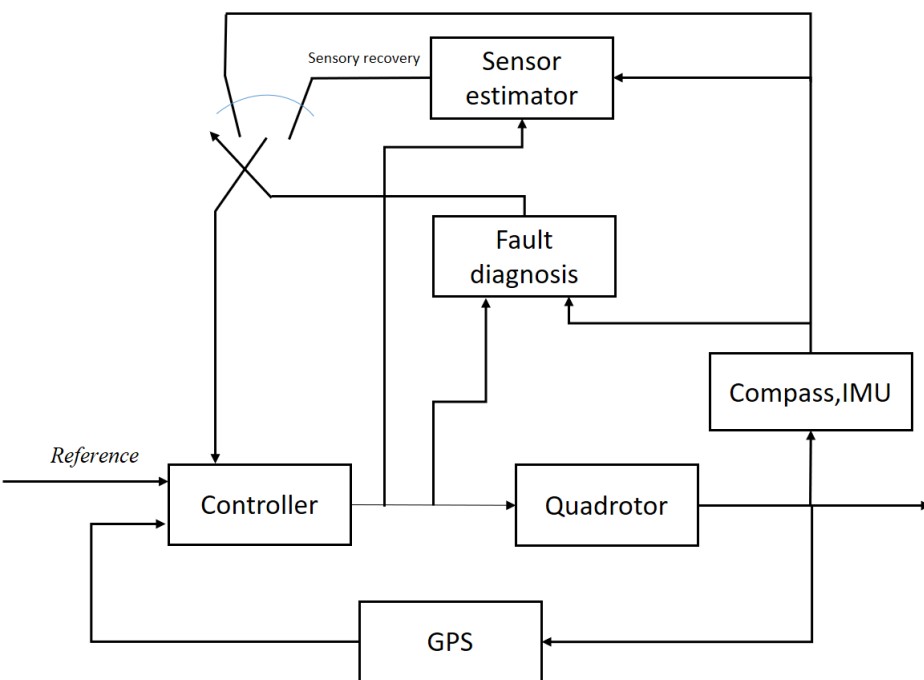

**Figure 7.** FTC against sensor fault with fault diagnosis.

## 5. Simulation Experiments

The proposed AI-based FTC can be verified by our simulation. During the simulation, the IMU sensor is assumed to have a fault at a certain time and the recovery is triggered to maintain the drone control. The simulation studies below use the computer ThinkPad X390 which has the 8th Gen Intel Core processors. MATLAB and its Deep Learning Toolbox are used in the simulation. The toolbox is a framework developed by the MathWorks used in the development of deep neural networks, including the LSTM network.

The rate gyro fault is considered. The proposed LSTM estimator can predict Euler angle and angular rate by using the previous states. The structure of the proposed LSTM network is composed of 10 inputs, 288 hidden units, and 6 outputs. For the LSTM learning, we use the random number to generate the four rotor control signals which are fed into the quadrotor model of [14] to receive the six output states (three Euler angles and three angular rates). In total, 59,960 sample data are collected. These data are divided into two groups: training set and test set. The training set has 50,981 samples, while the test set has 8979 samples. For time series $X_k, X_{k-1}, ...X_{k-D}$, the delayed term $D$ is 20. During the LSTM learning, it takes 158 epochs (about 592 min) satisfying the RMSE performance requirement. The convergence process is shown in Figure 8. For the training set, Figure 9 shows the estimates of Euler angles, while Figure 10 shows the estimates of angular rates (we zoom in the figure, having a better insight into the data). For the test set, Figure 11 shows the estimates of Euler angle, while Figure 12 shows the estimates of angular rate (we zoom in the figure, having a better insight into the data). It can be seen that the estimated values from the LSTM network closely match the sample data. This implies that the LSTM network can be used for estimating angular rate. Furthermore, the nonlinear estimator is designed by using (23) and (24), where the matrix $L$ is selected as

$$L = \begin{bmatrix} 0.9 & 0 & 0 \\ 0 & 0.9 & 0 \\ 0 & 0 & 0.9 \\ 0.9 & 0 & 0 \\ 0 & 0.9 & 0 \\ 0 & 0 & 0.9 \end{bmatrix} \tag{43}$$

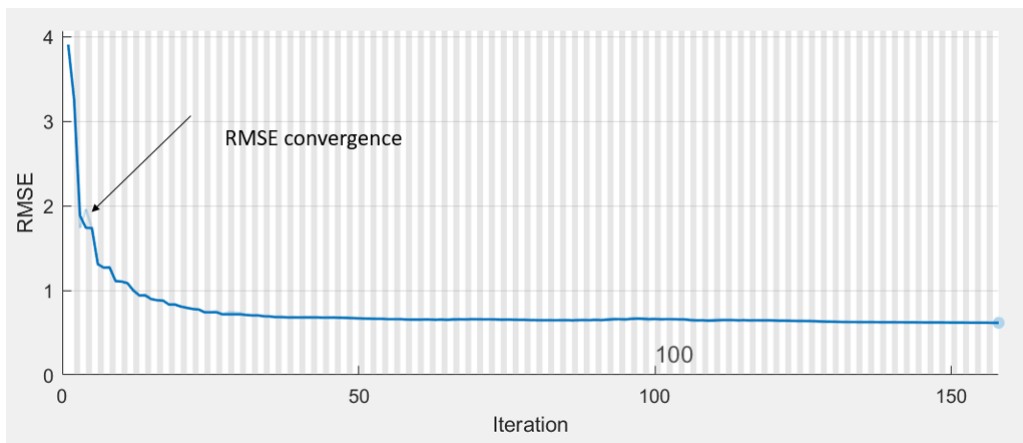

**Figure 8.** Convergence of the LSTM training.

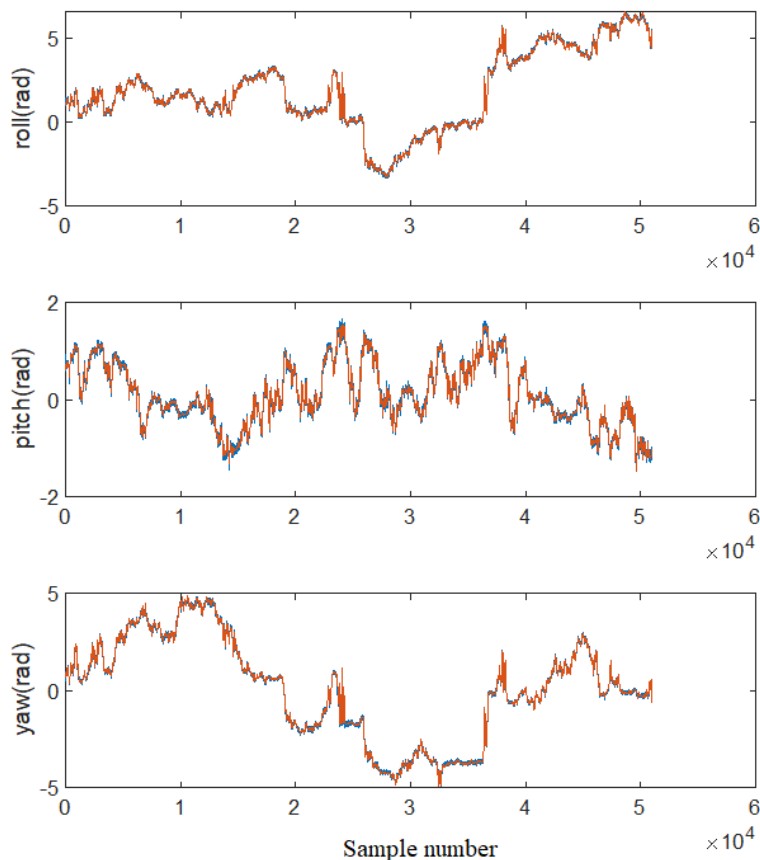

**Figure 9.** Estimated Euler angles of the LSTM training set: red line represents LSTM output; blue line represents sample output.

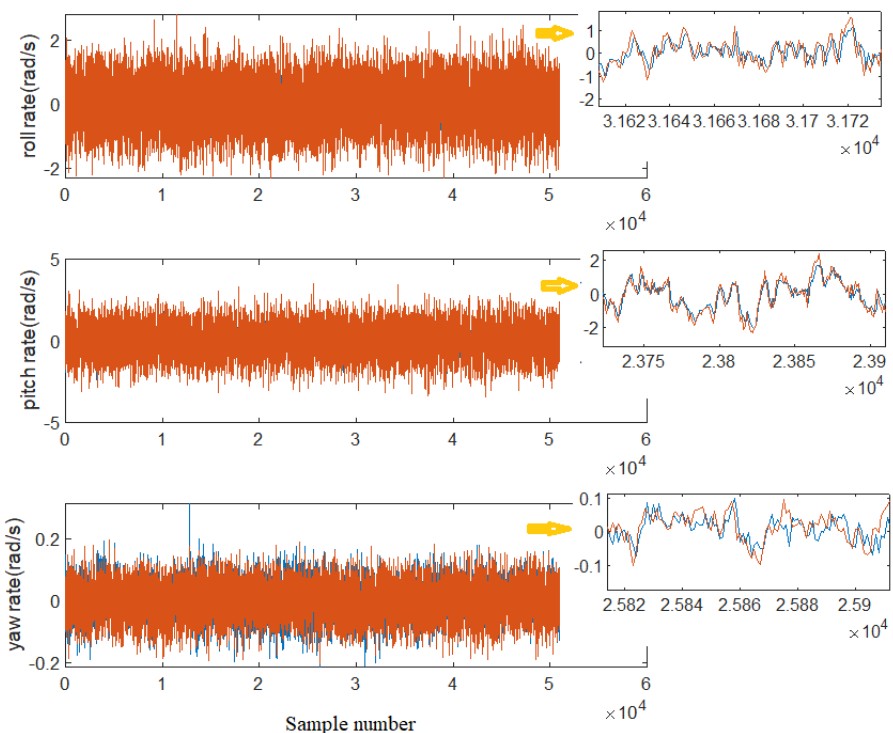

**Figure 10.** Estimated same angular rates of the LSTM training set: red line represents LSTM output; blue line represents sample output.

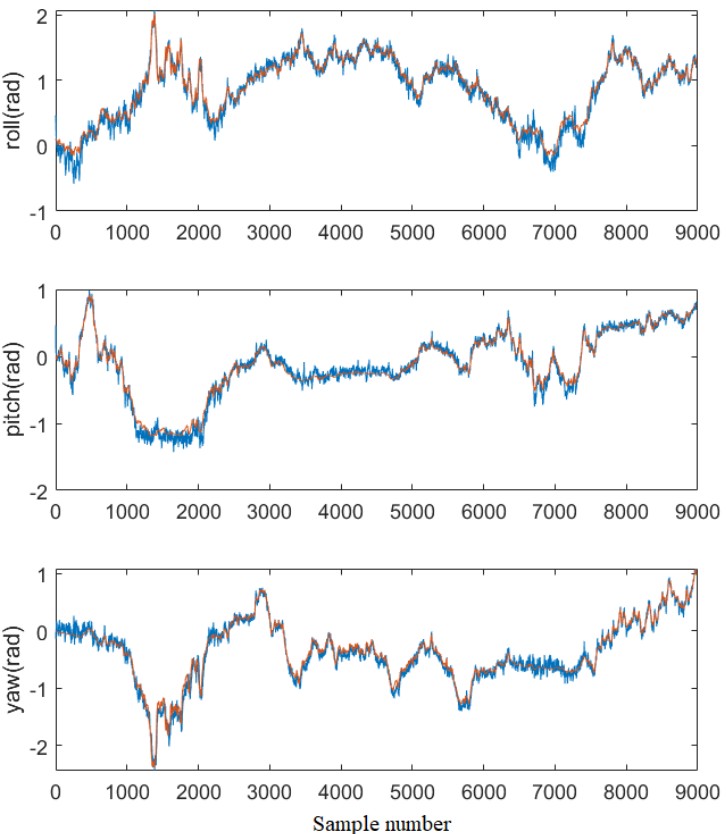

**Figure 11.** Estimated Euler angles of the LSTM test set: red line represents LSTM output; blue line represents sample output.

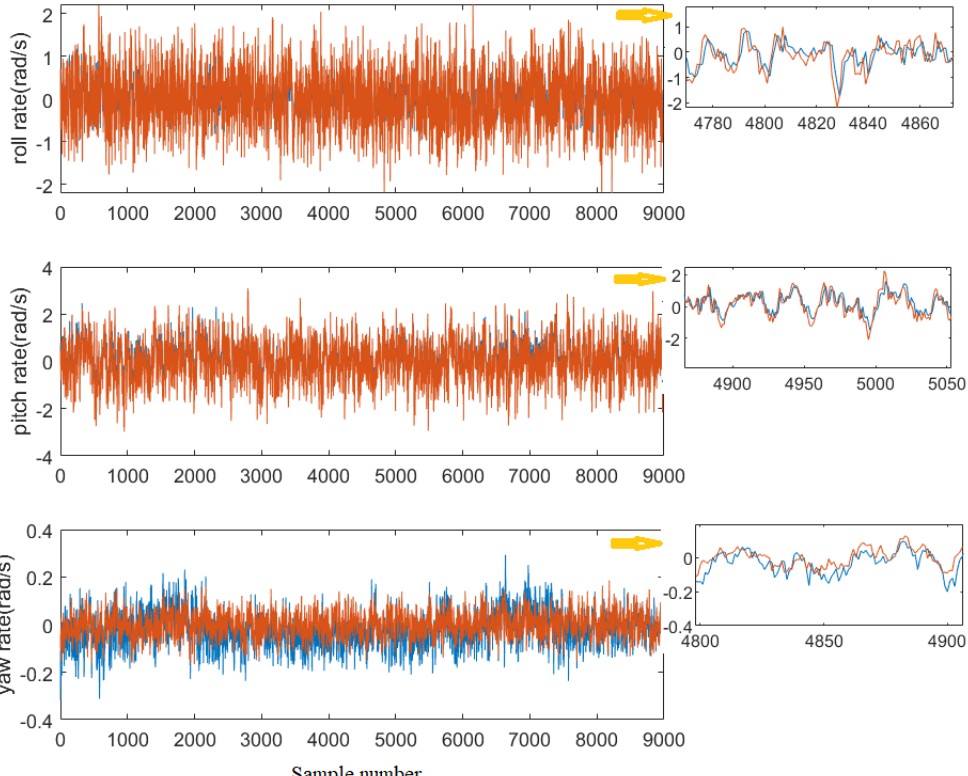

**Figure 12.** Estimated angular rates of the LSTM test set: red line represents LSTM output; blue line represents sample output.

As our discussion before, when having IMU fault, Euler angle estimator is necessary for calibration. The calibration rules are shown in Equations (40) and (41), where the factor $\alpha = 0.95$. The estimated Euler angles are input to the LSTM network.

In the simulation, it is assumed that a fault at IMU (roll rate) occurs at time = 25 s, where the bias fault is introduced (the bias offset is 1.2 rad/s). The random noise with amplitude = 0.00025 rad/s is added in the simulation. Figure 13 shows the position (xyz) result, while Figure 14 shows the 2D (xy) result. It should be noticed that the tracking performance along x, y, z axes is good, even the IMU sensor fault occurrence. For the fault diagnosis, we use Figure 15 to demonstrate it, where the threshold is chosen as 1.01 rad/s. It can be seen that the sensor fault is detected immediately once it happens. Thus, both Euler angle and LSTM estimators are used. The roll rate signal is recovered by using the proposed scheme. For Euler angle estimation, the result is shown in Figure 16. It can be seen that the difference between both the estimated and actual angles is small. For the roll and pitch rate estimation, the result is shown in Figure 17. From the figure, it is found that the outputs of the LSTM network are close to the actual ones without fault occurrence. We also tested the control without considering gyroscope fault estimator when the bias fault happens. In this case, we do not use any gyroscope sensor handling. Figure 18 shows the control performance of the position along x,y,z axes, while Figure 19 shows the 2D position profile along x,y axes. It is observed that the control performance is poor without considering gyroscope sensor handling. From the figures, it is shown that the position along y-axis is out of control. This also proves that a FTC technology has to incorporate sensor fault processing unit such that the system can continue to be operated safely.

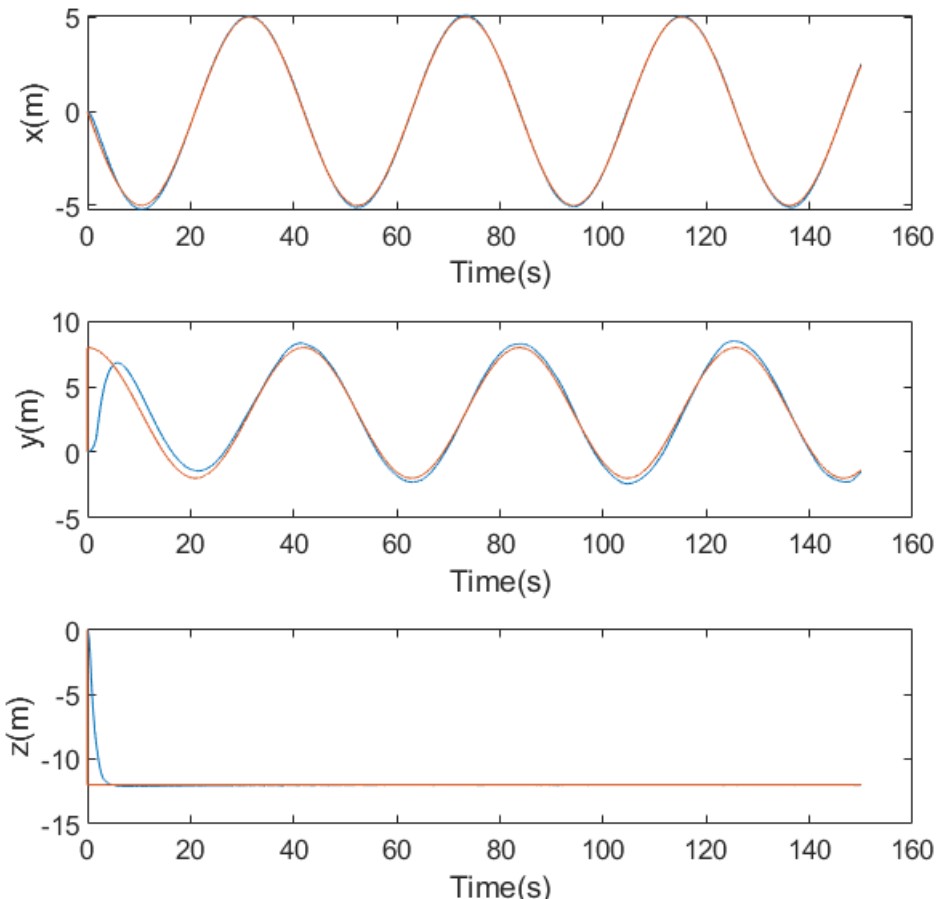

**Figure 13.** Bias fault: the position control performance (red line–desired, blue line–actual).

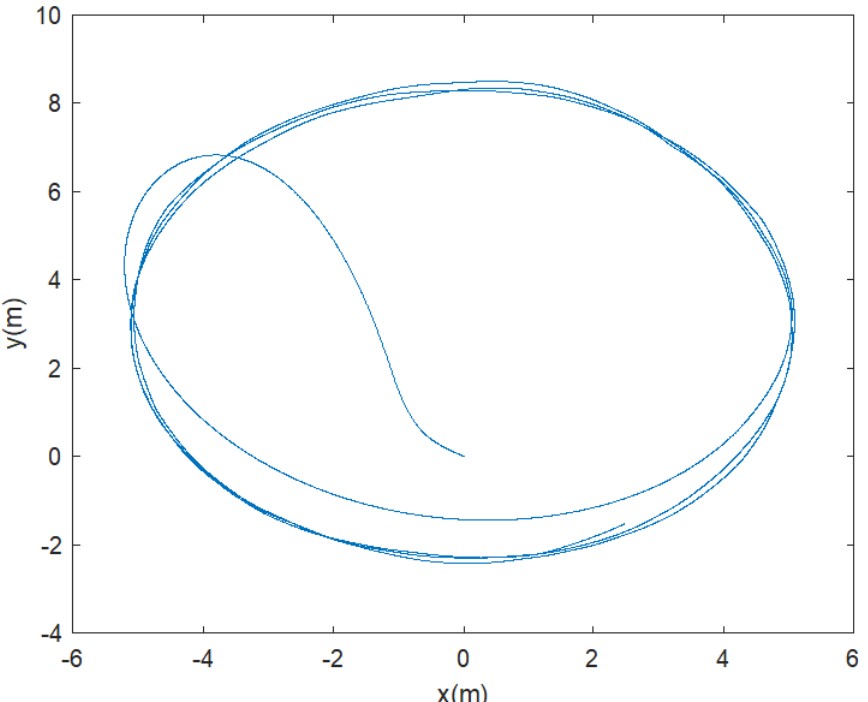

**Figure 14.** Bias fault: the 2D position control performance.

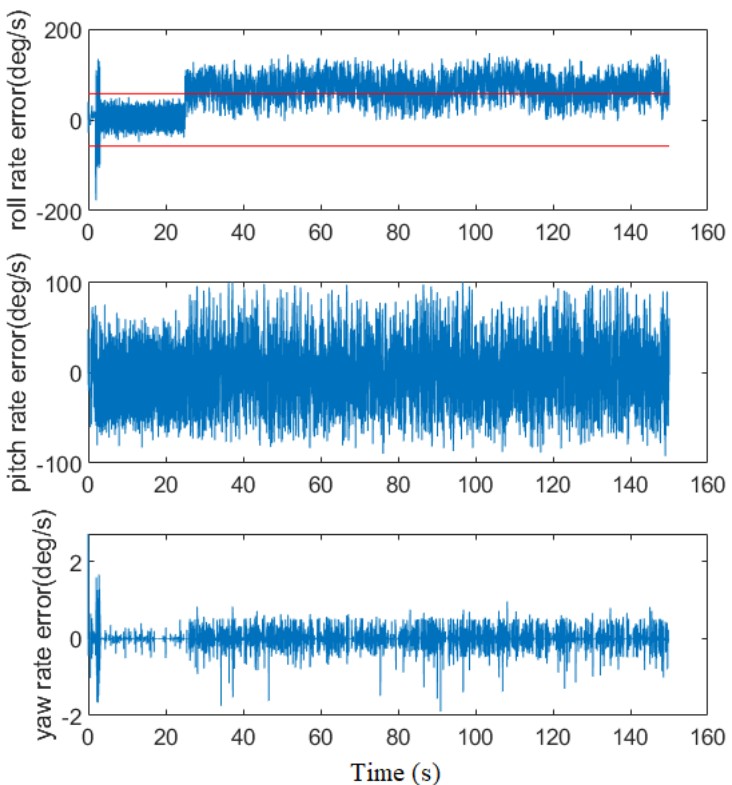

**Figure 15.** Bias fault: fault diagnosis result (red line–threshold, blue line–residual).

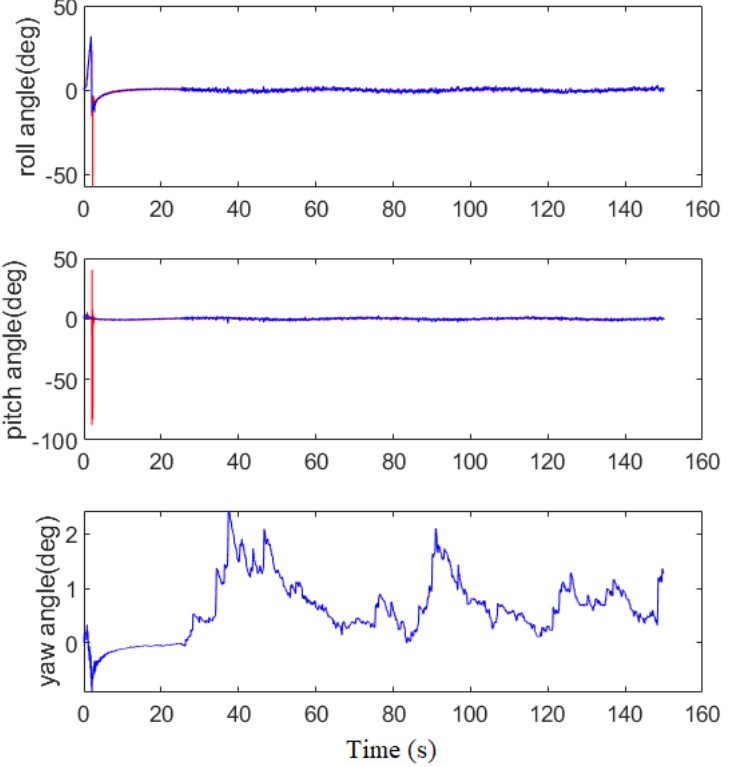

**Figure 16.** Bias fault: Euler angle estimation (red line–estimated angle, blue line–actual angle measurement).

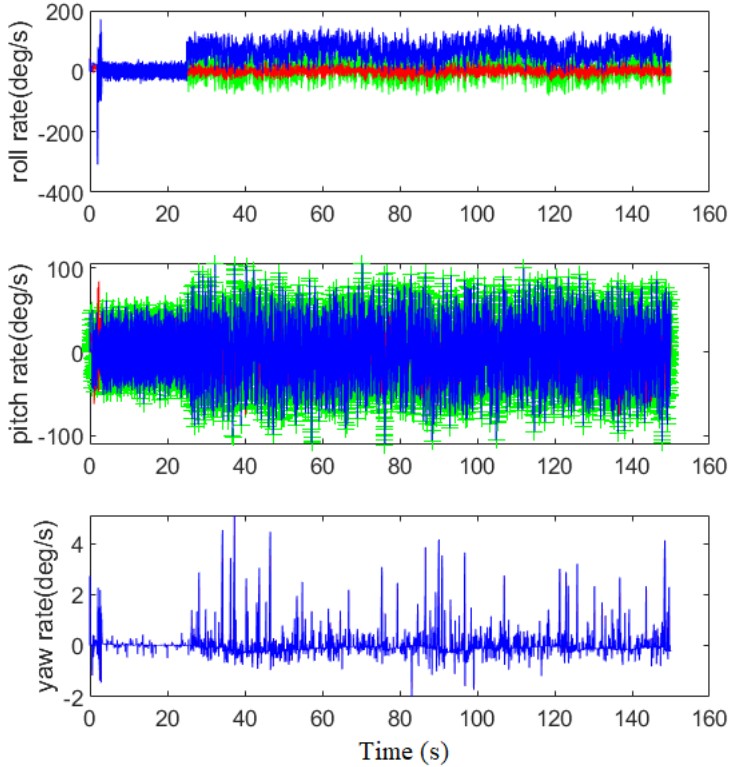

**Figure 17.** Bias fault: angular rate estimation (red line–estimated signal,blue line–faulty signal, green line–actual signal without fault occurrence).

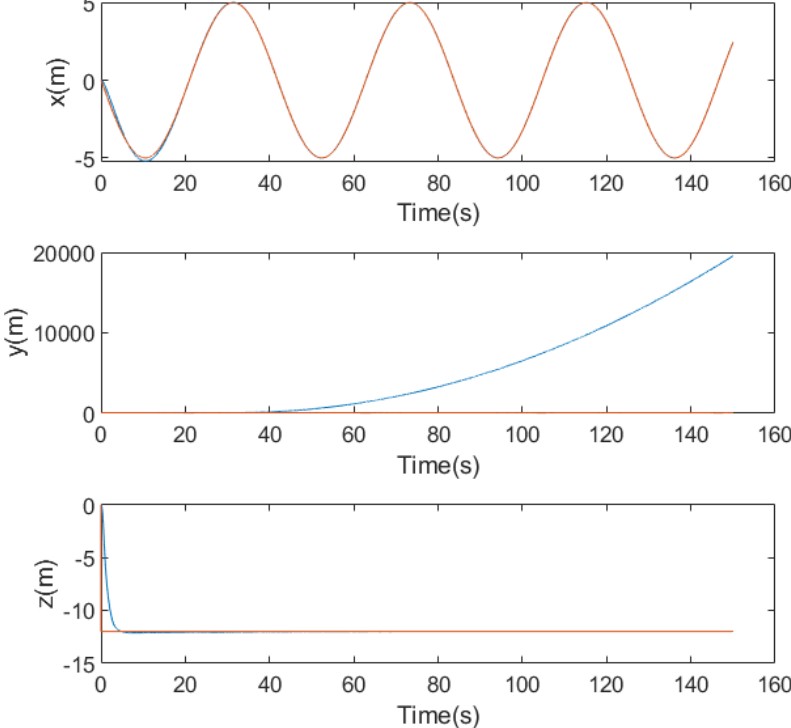

**Figure 18.** Bias fault: control result without sensor handling: the position profile (red line–desired, blue line–actual).

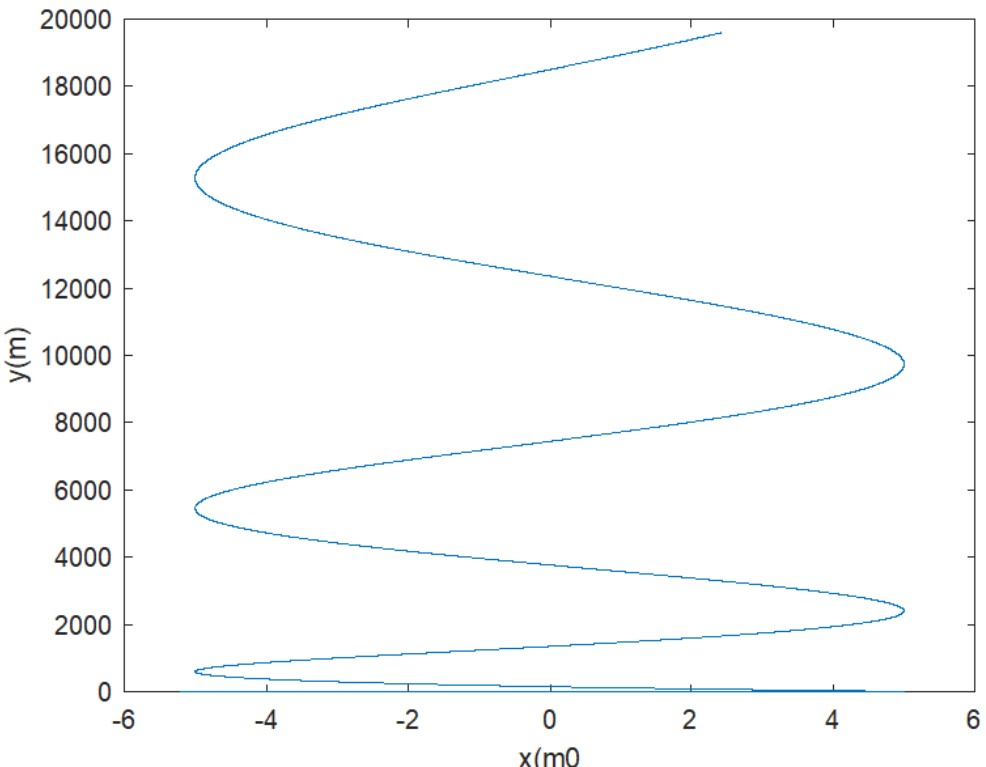

**Figure 19.** Bias fault: control result without sensor handling: the 2D position profile.

We consider another fault type—multiplication fault. In the simulation, it is assumed that the fault at IMU (roll rate) occurs at time = 25 s, where the roll rate multiplies by 2.8. For this kind of fault, Figure 20 shows the position (xyz) result, while Figure 21 shows the 2D (xy) result. It should be noticed that the tracking performance along x, y, z axes is still good, even with the fault occurrence. For the fault diagnosis, we use Figure 22 to demonstrate it, where the threshold is chosen as 1.01 rad/s. It can be seen that the sensor fault is detected immediately once it happens. Thus, both Euler angle and LSTM estimators are used. The roll rate signal is recovered by using the proposed scheme. For Euler angle estimation, the result is shown in Figure 23. It can be seen that the difference between both the estimated and actual angles is small. For the roll and pitch rate estimation, the result is shown in Figure 24. From the figure, it can be seen that the estimated angular rates are close to the actual ones. We also tested the control without considering the sensor estimator. In this case, we do not use any gyroscope sensor handling. Unfortunately, the attitude control is out of control at time = 27.28 s. Figure 25 shows the control result of Euler angle, where the maximum amplitudes of the roll, pitch and yaw angles are 79.25 rad (4557.3 deg), 33.02 rad (1891.9 deg) and $6.0074 \times 10^4$ rad ($3.442 \times 10^6$ deg) , respectively. Figure 26 shows the control result of angular rate. Obviously, a breakdown is observed from the angular rate control.

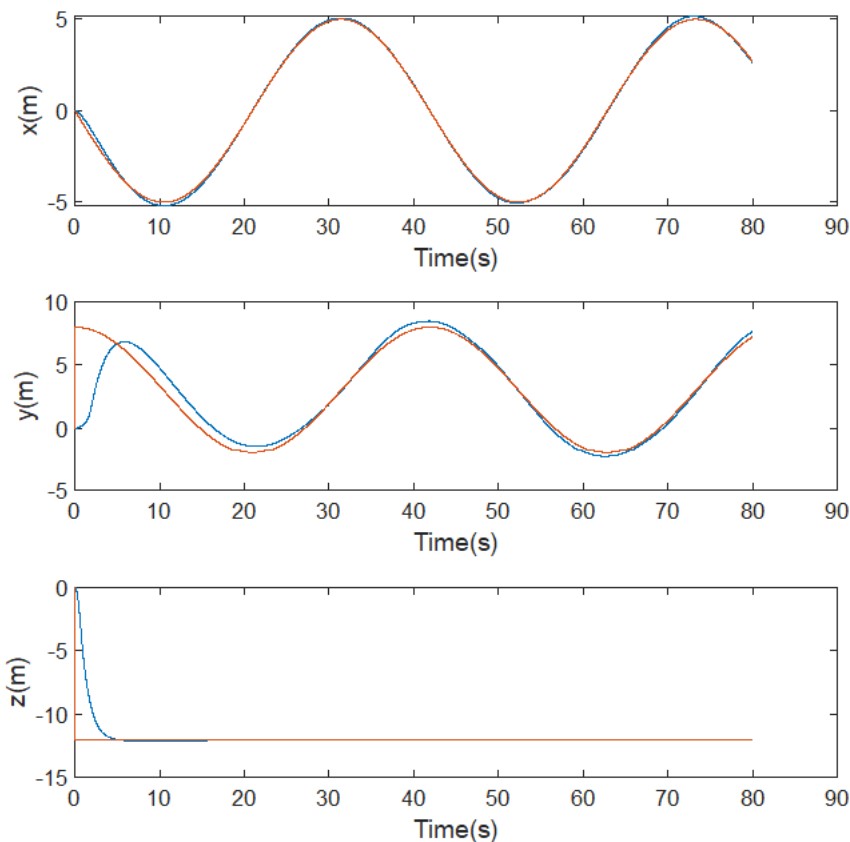

**Figure 20.** Multiplication fault : the position control performance (red line–desired, blue line–actual).

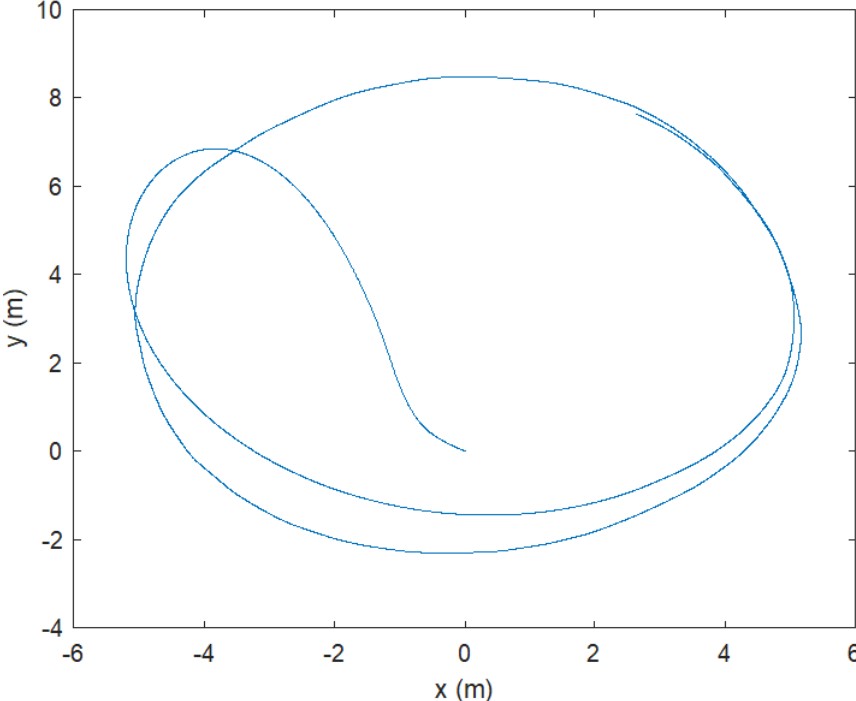

**Figure 21.** Multiplication fault: the 2D position control performance.

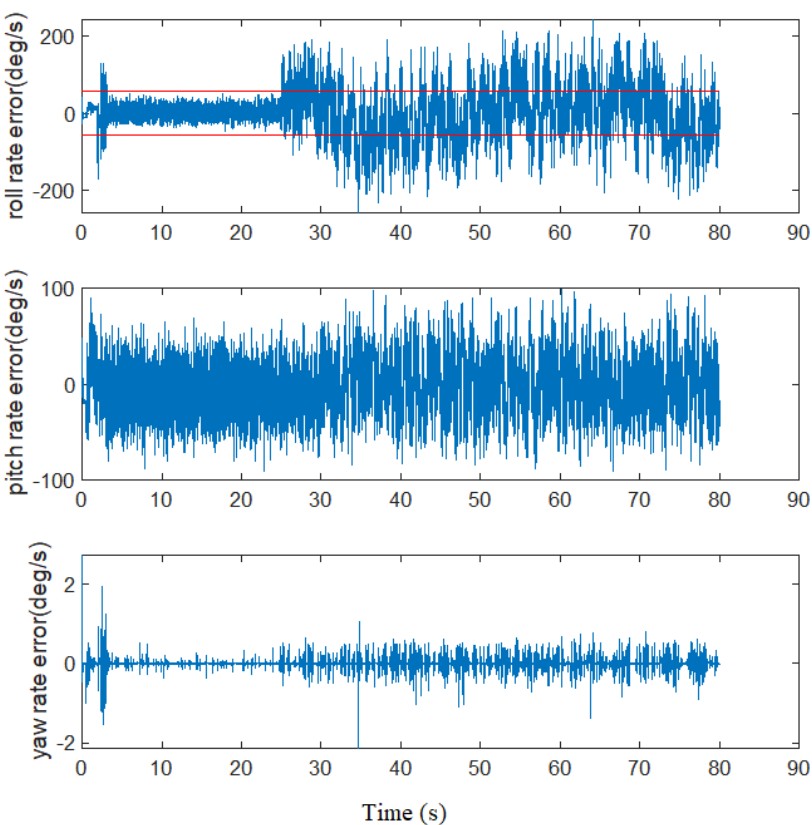

**Figure 22.** Multiplication fault: fault detection result (red line–threshold, blue line–residual).

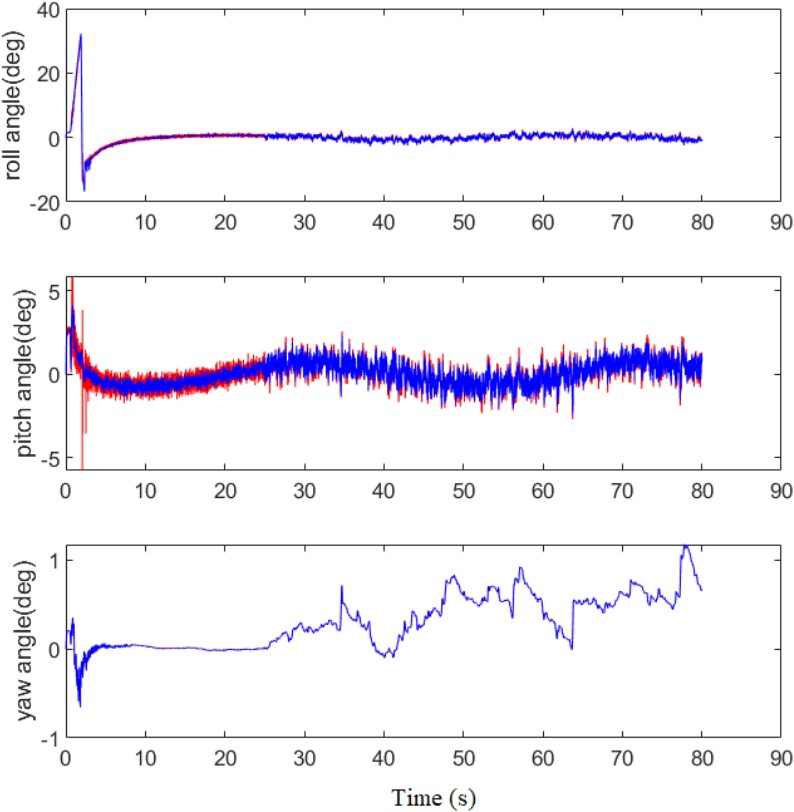

**Figure 23.** Multiplication fault: Euler angle estimation (red line–estimated angle, blue line–actual angle measurement).

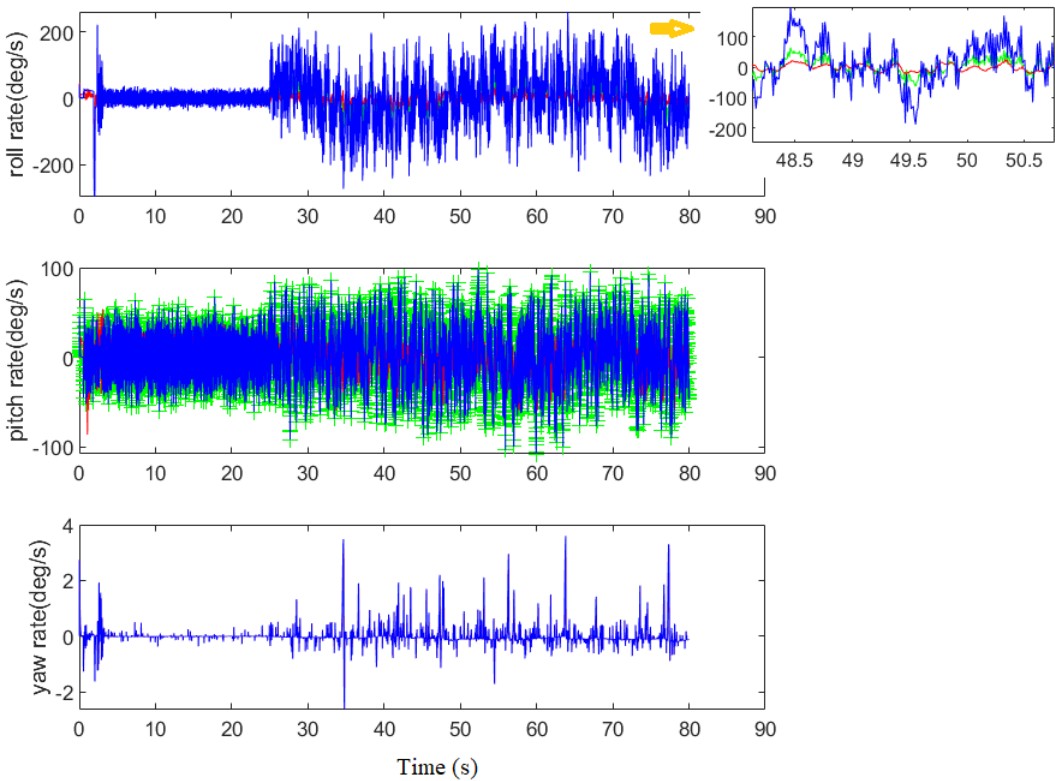

**Figure 24.** Multiplication fault: angular rate estimation (red line–estimated signal, blue line–faulty signal, green line–actual signal without fault occurrence).

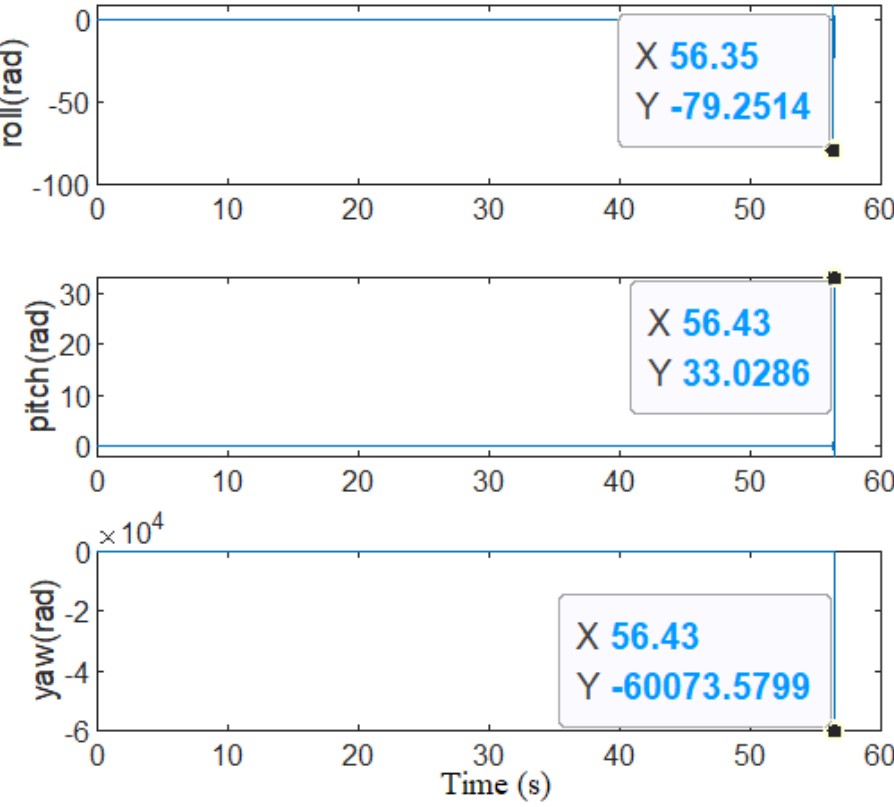

**Figure 25.** Multiplication fault: control result of Euler angle without sensor handling.

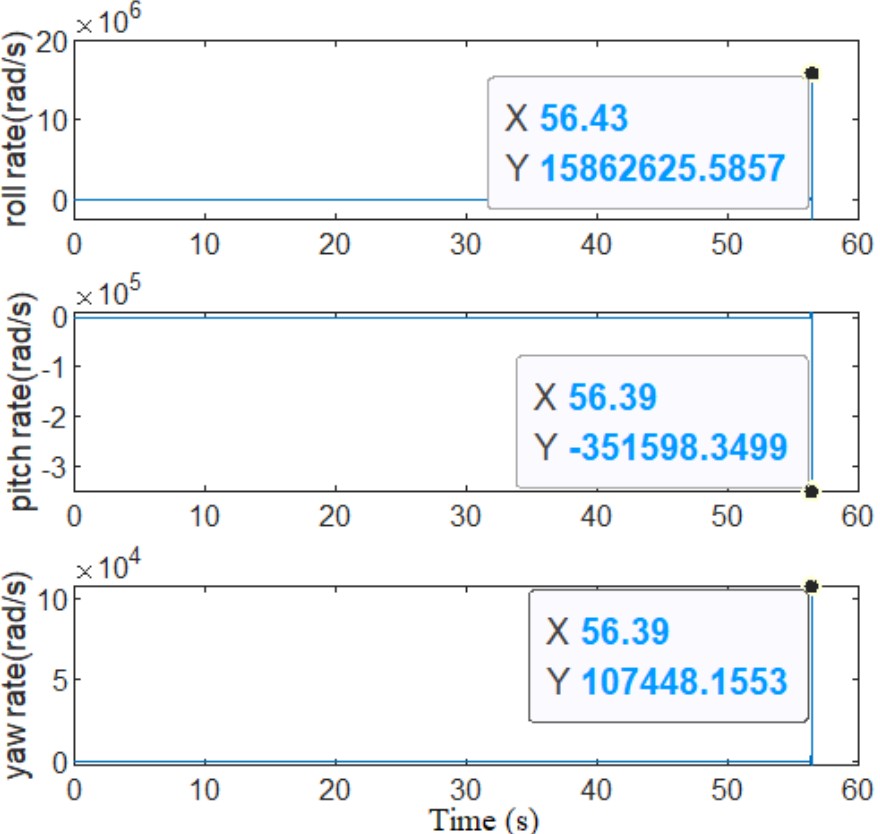

**Figure 26.** Multiplication fault: control result of angular rate without sensor handling.

The proposed LSTM estimator of the angular rate requires to learn sequence data: Euler angle, angular rate and all rotor control signals. As discussed in the previous section, the LSTM is a complex unit which uses a model based on short-term memory processes to build longer-term memory. Two study cases have shown that Euler angle and angular rate can be estimated correctly. Even if during the drone control process, the IMU fault occurs, the designed scheme still can diagnose the fault and recover the sensor information. The drawback of the proposed scheme is that during the whole recovery process, the other sensors such as accelerometer and compass must be healthy.

Comments: In [13], the authors use the LSTM for the IMU fault diagnosis and recovery. However, the authors assume that Euler angles are available. This is not realistic because the Euler angle measurement is a fusion of gyroscope with accelerometer. This implies that Euler angle measurement is not reliable in this situation. The proposed method solves this issue by adding the Euler angle estimator.

## 6. Discussion and Conclusions

The development of sensor fault diagnosis and recovery is an important topic in the drone control. AI-based design in estimating the angular rate of IMU is still a challenge. The difficulty point is that the estimator also involves Euler angle compensation due to the IMU fault.

This paper has presented a FTC design method when having a IMU sensor fault. It is assumed that the sensor fault may have a bias or multiplication fault. The proposed LSTM neural network is adopted to perform the supervised learning. Since the IMU fault also affects Euler angle measurement, we have proposed to use the translational equations for calibrating Euler angle. The detailed simulation has be given to show the effectiveness of the proposed method. For future work, we will implement the proposed algorithm in a real quadrotor. Furthermore, we will develop an anti GPS spoofing scheme for UAV control.

**Author Contributions:** Conceptualization: S.H.; methodology: S.H.; software: S.H. supervision: F.L., R.S.H.T. All authors have read and agreed to the published version of the manuscript.

**Funding:** This research received no external funding.

**Institutional Review Board Statement:** This paper has been approved by Temasek Lab. on 21 September 2022 (Number is 2752).

**Data Availability Statement:** The attached data set is for LSTM training.

**Conflicts of Interest:** The authors declare no conflict of interest.

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
