# Peer review of "Fault Tolerant Control of Quadrotor Based on Sensor Fault Diagnosis and Recovery Information"

_machines, doi:10.3390/machines10111088_

Round 1
Reviewer 1 Report
This paper investigates the fault-tolerant control method for quadrotor UAV based on sensor fault diagnosis and recovery information. However, there are some major issues that must be addressed:
1. The abstract and Introduction should be significantly revised to make the manuscript professional.
2. More existing results should analyzed.
3. The presentation is very bad, which makes the manuscript not readable. Please significantly adjust the manuscript presentation.
4. What is the contribution of the developed method? Please clarify this.
5. Comparisions between the proposed method and other existing methods should be conducted.
Author Response
This paper investigates the fault-tolerant control method for quadrotor UAV based on sensor fault diagnosis and recovery information. However, there are some major issues that must be addressed:
1. The abstract and Introduction should be significantly revised to make the manuscript professional.
Answer: In the revised version, the abstract and introduction have been revised properly.
2. More existing results should analyzed.
Answer: In the revised version, the existing results have been discussed.
3. The presentation is very bad, which makes the manuscript not readable. Please significantly adjust the manuscript presentation.
Answer: In the revised version, we revise the abstract and introduction and add simulation platform as well as the data collected.
4. What is the contribution of the developed method? Please clarify this.
Answer: In the revised version, the contributions are stated in the introduction section.
5. Comparsion between the proposed method and other existing methods should be conducted.
Answer: In the revised version, we add one Remark in the simulation section and compare the proposed method with the previous result.
Reviewer 2 Report
A fault-tolerant control to handle IMU sensor fault for quadrotor is proposed in this paper. Overall, the paper is well written and organized with a proper length. The contributions as well as the quality are both good. In addition, there are some points that are not very clear and should be addressed in the revised version:
1. The Introduction section is too short and 10 references are not enough for a regular research paper. What's more, please update the references, especially for recent years. For example, analytic model based approach is an important research issue which is widely applied on incipient fault diagnosis and fault tolerant control. The authors should supplement some results on this aspect, for example the following references had given significant design results:
[1] Incipient winding fault detection and diagnosis for squirrel-cage induction motors equipped
on CRH trains. ISA Transactions, 2020, 99: 488~495.
2. The description of the existing work should be shorter in Introduction section.
Furthermore, more descriptions of the proposed method are needed.
3. Pay attention to the quality of the figures. Take Figures 3-4 as example, the unified
fonts and improvement about image clarity are necessary. Please check the similar problems
in the paper.
4. Please add the necessary introduction of experiment platform and data acquiring.
Author Response
Q1. The Introduction section is too short and 10 references are not enough for a regular research paper. What's more, please update the references, especially for recent years. For example, analytic model based approach is an important research issue which is widely applied on incipient fault diagnosis and fault tolerant control. The authors should supplement some results on this aspect, for example the following references had given significant design results:
[1] Incipient winding fault detection and diagnosis for squirrel-cage induction motors equipped
on CRH trains. ISA Transactions, 2020, 99: 488~495.
Answer: In the revised version, the introduction is re-written and the fault diagnosis is also discussed. [1] has been cited. Totally, 15 references are used in the revised version.
Q2. The description of the existing work should be shorter in Introduction section. Furthermore, more descriptions of the proposed method are needed.
Answer: In the revised version, we shorten the more contents regarding existing works and add more explanations of the proposed method.
Q3. Pay attention to the quality of the figures. Take Figures 3-4 as example, the unified fonts and improvement about image clarity are necessary. Please check the similar problems in the paper.
Answer: In the revised version, Figures 3 and 4 have been improved. We also improved the quality of the other figures.
4. Please add the necessary introduction of experiment platform and data acquiring.
Answer: In the revised version, the platform (hardware and MATLAB) and data collected have been added. The collected 60000 data (used neural network learning) are attached with the paper (if the paper is published, the data is open).
Reviewer 3 Report
Review
The paper is well written and results are good.
Proposition for the improvement of the paper:
There could be more common references to FDD topic. For example, the following FTC article has a similar fault detection method:
J Kortela, SL Jämsä-Jounela. 2015. Fault-tolerant model predictive control (FTMPC) for the BioGrate boiler. 2015 IEEE 20th Conference on Emerging Technologies & Factory Automation.
At least 5/10 of the references have a same author as the first writer. Proposition: try to find references to other similar papers.
Figures
Please improve the figures quality:
The resolution of the pictures should be at least 600 pixels / inch
Small errors
In row 69 there is a small spelling error “all ?nite”
In row 70, d could be italic “dG”
In row 84, two d:s could be italic “dG”
In row 114, there should space after “:” -> “parts: input”
In row 138 “we re-arranging” should be “we re-arrange”
In row 214, there should be space after Eqs “Eqs40-41”
Author Response
The paper is well written and results are good.
Proposition for the improvement of the paper:
Q1.There could be more common references to FDD topic. For example, the following FTC article has a similar fault detection method:
J Kortela, SL Jämsä-Jounela. 2015. Fault-tolerant model predictive control (FTMPC) for the BioGrate boiler. 2015 IEEE 20th Conference on Emerging Technologies & Factory Automation.
Answer: In the revised version, we add the six related FDD papers, including the recommended article.
Q2.At least 5/10 of the references have a same author as the first writer. Proposition: try to find references to other similar papers.
Answer: In the revised version, we remove one paper of the first author and add 6 more papers to the references in the paper. The ratio between the first author and the other references is 4/15.
Q3. Figures
Please improve the figures quality:
The resolution of the pictures should be at least 600 pixels / inch
Answer: In the revised version, the quality of the figures such as Figures 2-4, is improved.
Q4. Small errors
In row 69 there is a small spelling error “all ?nite”
In row 70, d could be italic “dG”
In row 84, two d:s could be italic “dG”
In row 114, there should space after “:” -> “parts: input”
In row 138 “we re-arranging” should be “we re-arrange”
In row 214, there should be space after Eqs “Eqs40-41”
Answer: In the revised version, these errors have been corrected.
Round 2
Reviewer 1 Report
Based on the evaluation of the submitted files, I would like to give accept decision
Reviewer 2 Report
In a general way most of my comments were answered by the authors. My overall opinion about this paper is quite good. The manuscript is well written and acceptable for publishing